# Primary cilia control cellular patterning of Meibomian glands during morphogenesis but not lipid composition

Céline Portal [1,6], Yvonne Lin[1,7], Varuni Rastogi[1,7], Cornelia Peterson [2], Samuel Chi-Hung Yiu[1], James W. Foster[1], Amber Wilkerson [3], Igor A. Butovich[3,4] & Carlo Iomini [1,5 ✉]

Meibomian glands (MGs) are modified sebaceous glands producing the tear film's lipids. Despite their critical role in maintaining clear vision, the mechanisms underlying MG morphogenesis in development and disease remain obscure. Cilia-mediate signals are critical for the development of skin adnexa, including sebaceous glands. Thus, we investigated the role of cilia in MG morphogenesis during development. Most cells were ciliated during early MG development, followed by cilia disassembly during differentiation. In mature glands, ciliated cells were primarily restricted to the basal layer of the proximal gland central duct. Cilia ablation in keratine14-expressing tissue disrupted the accumulation of proliferative cells at the distal tip but did not affect the overall rate of proliferation or apoptosis. Moreover, impaired cellular patterning during elongation resulted in hypertrophy of mature MGs with increased meibum volume without altering its lipid composition. Thus, cilia signaling networks provide a new platform to design therapeutic treatments for MG dysfunction.

[1] Department of Ophthalmology, Wilmer Eye Institute, Johns Hopkins University School of Medicine, Baltimore, MD 21231, USA. [2] Department of Molecular and Comparative Pathobiology, Johns Hopkins University School of Medicine, Baltimore, MD 21205, USA. [3] Department of Ophthalmology, University of Texas Southwestern Medical Center, Dallas, TX 75390, USA. [4] Graduate School of Biomedical Sciences, University of Texas Southwestern Medical Center, Dallas, TX 75390, USA. [5] Department of Cell Biology, Johns Hopkins University School of Medicine, Baltimore, MD 21231, USA. [6] Present address: Sorbonne Université, INSERM, CNRS, Institut de la Vision, 17 rue Moreau, F-75012 Paris, France. [7] These authors contributed equally: Yvonne Lin, Varuni Rastogi. ✉email: ciomini1@jhmi.edu

Meibomian glands (MGs) are holocrine glands located within the tarsal plates of the upper and lower eyelids. These modified sebaceous glands (SGs) are composed of clusters of secretory acini that discharge their secretion into several shorter ductules branching from the central duct of the gland. The secretory product, meibum (composed of lipids, proteins and nucleic acids of the whole cell), is ultimately released at the eyelid margin. The meibum subsequently spreads onto the ocular surface, as the outermost layer of the tear film, with each blink[1,2]. This lipid-rich layer plays crucial protective roles for the ocular surface, as it functions as a lubricant for the eyelids during blinking, prevents tear overflow onto the lids, and reduces tear evaporation[1,3].

Defective MGs lead to meibomian gland dysfunction (MGD), defined as "a chronic, diffuse abnormality of the MGs, commonly characterized by terminal duct obstruction and/or qualitative/quantitative changes in the glandular secretion"[4]. Reduced lipid secretion may contribute to tear film instability and facilitate entry into the vicious cycle of dry eye disease (DED), among the most commonly encountered ophthalmic diseases with a global prevalence ranging from 5 to 50%[5]. DED is subdivided into two primary and non-mutually exclusive categories: aqueous deficient dry eye (ADDE) and evaporative dry eye (EDE)[6]. MGD is considered the leading cause of EDE and DED[6–8]. The recent development of therapeutic solutions targeting MGD includes ocular lubricants, eyelid-warming devices and intense pulsed light, primarily focusing on relieving MG obstruction or replacing lipids[9]. However, there is an unmet need for treatments to prevent MG atrophy and stimulate lipid production. This deficiency in current effective pharmacological targets is due primarily to the very limited knowledge about molecular networks underlying MG development and renewal that could be the target for an efficient and long-lasting treatment of MGD.

Human MG formation occurs during embryonic development between the third and the seventh month of gestation, corresponding to the sealed-lid phase of eyelid development[1]. In mice, MG development begins at embryonic day 18.5 (E18.5) and continues postnatally[10]. As in humans, MG development in mice occurs during the sealed-lid phase of eyelid development, which is indispensable for MG development[11,12].

Although it has been suggested that MG development shares similarities with the development of the pilosebaceous unit comprising a hair follicle and its associated SGs, the basic mechanisms underlying MG development and renewal remain poorly understood. Like hair follicles, MGs develop from the ectodermal sheet, which invaginates into the mesoderm to form an anlage. Then, similar to the hair anlage of eyelashes, the meibomian anlage develops lateral outgrowths that later differentiate into ductules and sebaceous acini[13]. In murine development, an epithelial placode forms at E18.5, followed by invagination in the mesenchyme and elongation of the placode, branching of the MGs beginning around postnatal day 5 (P5), and acquisition of their mature morphology by P15[10].

Primary cilia are microtubule-based cellular organelles that originate from the basal body and extend from the plasma membrane. Intraflagellar transport (IFT), a bidirectional movement of protein particles along the axoneme, ensures the appropriate assembly and maintenance of cilia[14–18]. Dysfunction of the primary cilium produces a heterogeneous group of diseases termed ciliopathies, some of which induce severe developmental defects, highlighting the crucial role of the primary cilium in tissue development[19]. The primary cilium plays essential roles in the development of ectoderm-derived tissues, including the skin, the corneal epithelium, and the pilosebaceous unit[20,21]. In particular, the primary cilium modulates corneal epithelial thickening through the regulation of cell proliferation and vertical migration[22]. In the skin, primary cilia limit the hyperproliferation of keratinocytes in the epidermis[23,24].

Moreover, primary cilia ablation leads to hair follicle morphogenesis arrest[24–30] and dysregulation of the hair growth cycle[31]. Patients affected by Bardet-Biedl syndrome, an autosomal recessive ciliopathy, suffer from several cutaneous conditions, including keratosis pilaris and seborrheic dermatitis[32]. Interestingly, primary cilia ablation induces hyperplasia of SG lobules, indicating a cilia-dependent regulatory role in SG development[23]. However, the pathogenesis of these cilia-associated cutaneous conditions and the mechanisms underlying the abnormal enlargement of SGs remain unknown. Although the role of the primary cilium has been investigated in various ectoderm-derived tissues, its role in MG development, maintenance and function remains unknown.

In this study, we show that MG cells are ciliated in the early stages of development, and meibocytes lose their primary cilium as they differentiate. We demonstrate that the primary cilium is required for regulating the central duct diameter and overall size of MGs. We propose a mechanism by which primary cilia determine the early MG cell patterning by controlling the spatial distribution of proliferating and dying cells within developing glands. These findings suggest cilia-mediated signaling pathways as potential therapeutic targets to counteract MGD.

## Results

**Developmental loss of primary cilia in K14-expressing tissues leads to an abnormal increase in MGs size and lipid content.** To determine the involvement of primary cilia in MG morphogenesis, we generated the conditional knockout K14-Cre;Ift88[fl/fl] (here referred to as cKO). In this mouse, the *Ift88* gene, encoding for a subunit of the IFT machinery required for cilia assembly and maintenance[33], is excised in all epithelial cells expressing Keratin 14 (K14), including MG tissues. The expression of the *K14-Cre* recombinase was followed by using the mT/mG reporter mouse line[34] (Supplement Fig. 1). In the *K14-Cre;Ift88[floxed];mT/mG* transgenic line, the Cre-dependent excision of a cassette expressing the red-fluorescent membrane-targeted tdTomato (mT) drove the expression of a membrane-targeted green fluorescent protein (mG) in K14-expressing tissues (Supplement Fig. 1). To monitor primary cilia ablation in MGs, we immunodetected ARL13B, a protein associated with the ciliary membrane[35,36]. At P3, cilia were present on virtually all MG cells of the control mice. In contrast, cilia were absent or very short in MG cells of cKO mice (Supplement Fig. 1). As previously described, the external appearance of newborn cKO mice was generally similar to that of control mice[22,23]. Because defects in eyelid fusion and opening during development can affect MG morphogenesis, we analyzed in detail these processes[11]. Eyelid fusion and opening in both cKO and control mice occurred around E15.5 and P13, respectively, confirming the finding of previous studies[22,23]. However, we noticed the presence of multifocal, white, granular to chalky seborrheic debris along the eyelid margins of most adult cKO mice, which was not observed in control mice (Fig. 1a).

Oil red O (ORO) staining of the tarsal plate revealed that the number of MGs per eyelid (upper and lower) was similar in control and cKO mice at P6 and P8 (Fig. 1b, c). However, stained areas of the cKO MGs were 29% and 21% larger than those of control mice at P6 and P8, respectively (Fig. 1c). Mature MGs, as seen in mice at P21, appeared very close to each other in both cKO and control mice, challenging the accurate outlining of individual glands and thus the measurement of their surface (Fig. 1d). However, MGs appeared more densely distributed in the tarsal plates of the cKO than in those of control mice. Unstained foci were visible between the proximal region of adjacent MGs in control, but not in the cKO (arrows in Fig. 1d). Moreover, no significant differences in MG dimensions were observed between males and females of the same genotype (Supplement Fig. 2), and both sexes

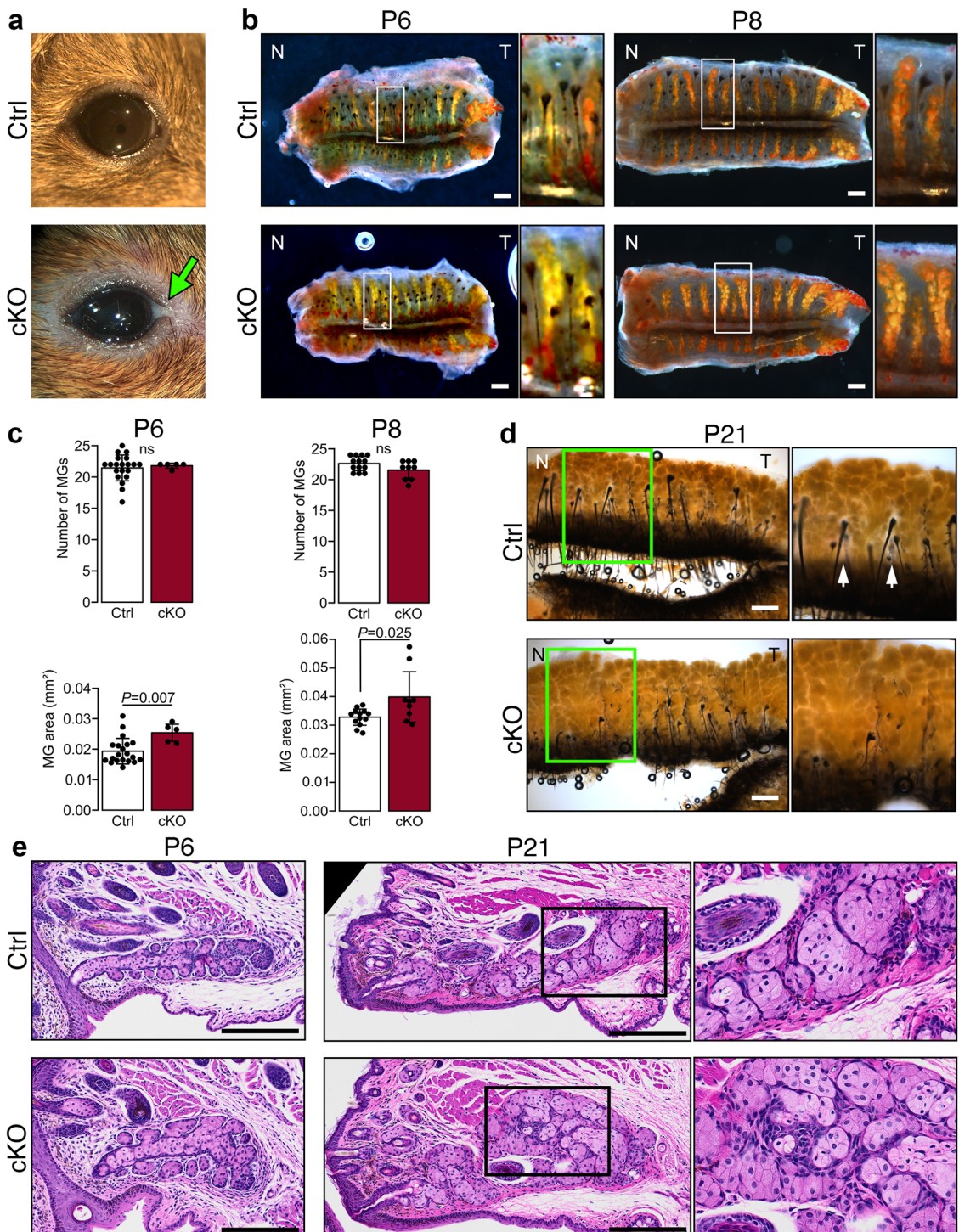

**Fig. 1 Primary cilium ablation leads to larger MGs. a** Representative pictures of control and cKO adult (6 months) eyes. Arrow indicates a white deposit, only observed in cKO mice. **b** Representative images of tarsal plates stained with ORO at P6 and P8. Boxed regions indicate the areas shown at higher magnification. Scale bar, 200 µm; N, nasal; T, temporal. **c** The number of MGs and MG size were quantified at P6 and P8 ($n = 20$ controls and 5 cKO mice at P6; $n = 13$ controls and 9 cKO mice at P8). Per mouse, MG area was determined by averaging the MG area of all individual MGs in the upper and lower eyelids. Data were presented as mean ± SD. Statistical significance was assessed using the Mann-Whitney test. ns, nonsignificant, $P \geq 0.05$. **d** Representative images of tarsal plates stained with ORO at P21. Boxed regions indicate the areas shown at higher magnification. Scale bar, 200 µm; N, nasal; T, temporal. **e** Representative images of control and cKO MGs stained with HE at P6 and P21. Boxed regions indicate the areas shown at higher magnification. Scale bar, 200 µm.

were pooled for the entire study. Despite differences in MG size, histological analysis revealed that the morphology of basal and differentiating meibocytes was similar in both mutant and control, and no sign of duct obstruction was observed in mutant mice (Fig. 1e).

Given the significant expansion of the mutant MGs, we asked whether the lipid production and composition were altered in the cilia mutant. Lipid profiles of the tarsal plate extracts were assessed by high-resolution mass spectrometry (MS) in combination with isocratic and gradient reverse-phase ultra-high performance liquid

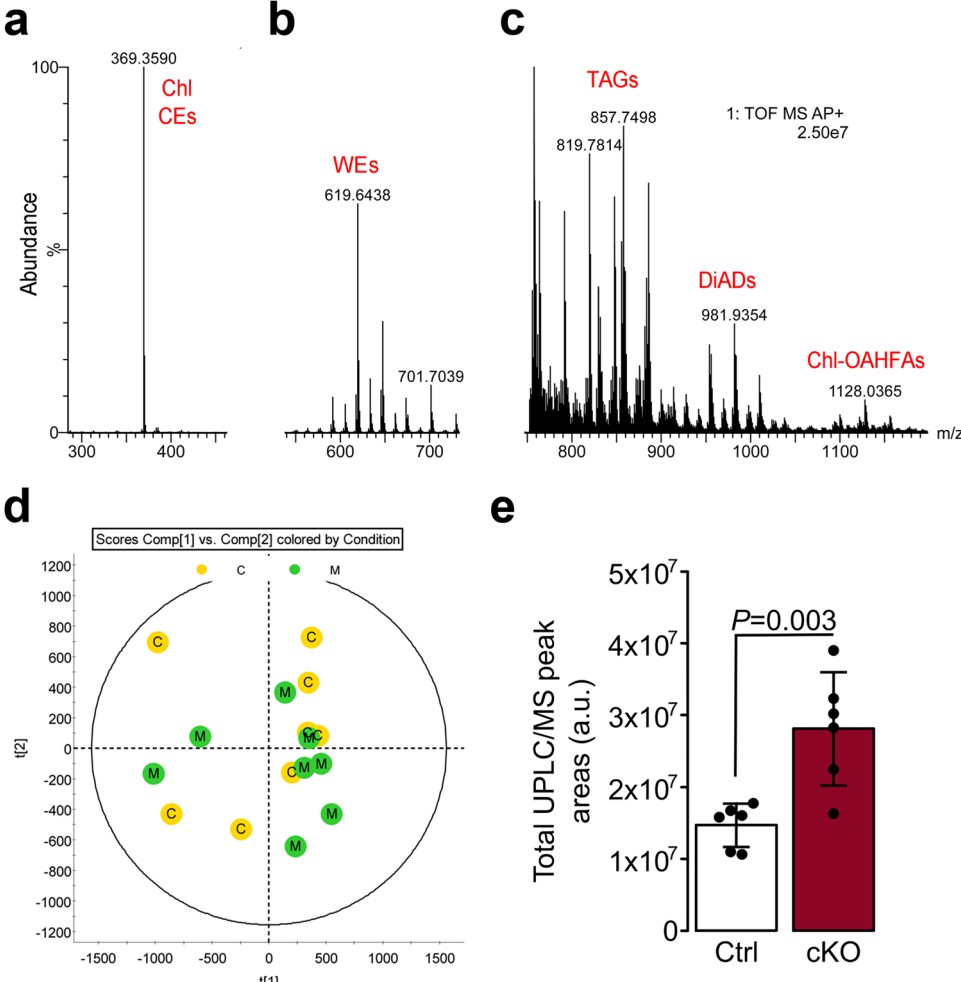

**Fig. 2 Reverse phase liquid chromatography/high-resolution time of flight atmospheric pressure chemical ionization mass spectrometry (LC/MS) analysis of mouse meibomian lipids conducted in the positive ion mode revealed close similarity in the lipidomes of control and cKO mice, but a two-fold increase in the total lipid content of the latter. a** An analytical signal of free cholesterol (Chl) and cholesteryl esters (CEs) from a control mouse. **b** An observation spectrum of the pool of meibomian wax esters (WEs) from a control mouse. (**c**) An observation spectrum of the pools of triacylglycerols (TAGs), α,ω-diacylated diols (DiADs), and cholesteryl esters of (O)-acylated ω-hydroxy fatty acids (Chl-OAHFAs) from a control mouse. **d** A scores plot generated using Principal Component Analysis (PCA) for the control (C, yellow dots) and cKO (M, green dots) LC/MS data demonstrated a strong overlap of the control and mutant samples of meibomian lipids with no clear clustering of the samples of different types, indicating their close biochemical compositions. **e** However, primary cilium ablation led to a higher overall lipid production in the tarsal plates of cKO mice compared with the lipid content of control mice ($n = 6$ for Ctrl and $n = 6$ for cKO). Data were presented as mean ± SD. Statistical significance was assessed using the Mann-Whitney test.

chromatography. Approximately 150 analytes with unique combinations of retention times and mass-to-charge ($m/z$) ratios were identified. Representative mass spectra of a control wild-type sample are shown in Fig. 2a–c. The principal component analysis (PCA) produced no obvious clustering of the control or cKO samples (Fig. 2d), implying that their chemical compositions were similar. However, analysis of samples for a set of 15 major lipid species from the wax ester (WE) and the cholesteryl ester (CE) families showed approximately a two-fold increase in the amount of produced lipids in the cKO mutant compared to control mice (Fig. 2e). Collectively, primary cilia ablation in the MGs led to the expansion of MGs size and a two-fold increase in lipid production without affecting the overall maturation process of meibocytes and, therefore, the lipid composition of the meibum.

**Primary cilia progressively disassemble during MG development and meibocyte maturation.** To gain insights into the role of cilia in MG size control, we sought to determine the spatio-temporal distribution of primary cilia in the developing and

mature MGs of the Arl13b-mCherry;Centrin2-GFP transgenic mouse (Fig. 3). This mouse expresses ARL13B, a cilia-membrane-associated protein, fused to the monomeric red fluorescent protein mCherry, and Centrin2, a centriolar protein, fused to GFP resulting in red and green fluorescently-labeled cilia and basal bodies, respectively[35–38]. Because the size and morphology of MGs varies according to their position within the tarsal plate (Fig. 1b), we numbered MGs starting from the biggest gland at the temporal side of the eyelid (Fig. 3a). Throughout this study we analyzed glands preferentially located at a similar middle position in the tarsal plate, as illustrated in Fig. 3a, b, in mutant and control mice.

At P3, more than 70% of MG cells displayed primary cilia emanating from the apical side of basal cells into the center of the developing gland (Fig. 3c, d, g). A similar proportion of MG cells were ciliated during early MG branching at P6 and P8 (Fig. 3e, g). At P6, primary cilia were visible in the elongating distal tip, the developing duct, and the budding acini; however, basal bodies of mature meibocytes located at the center of the central duct did

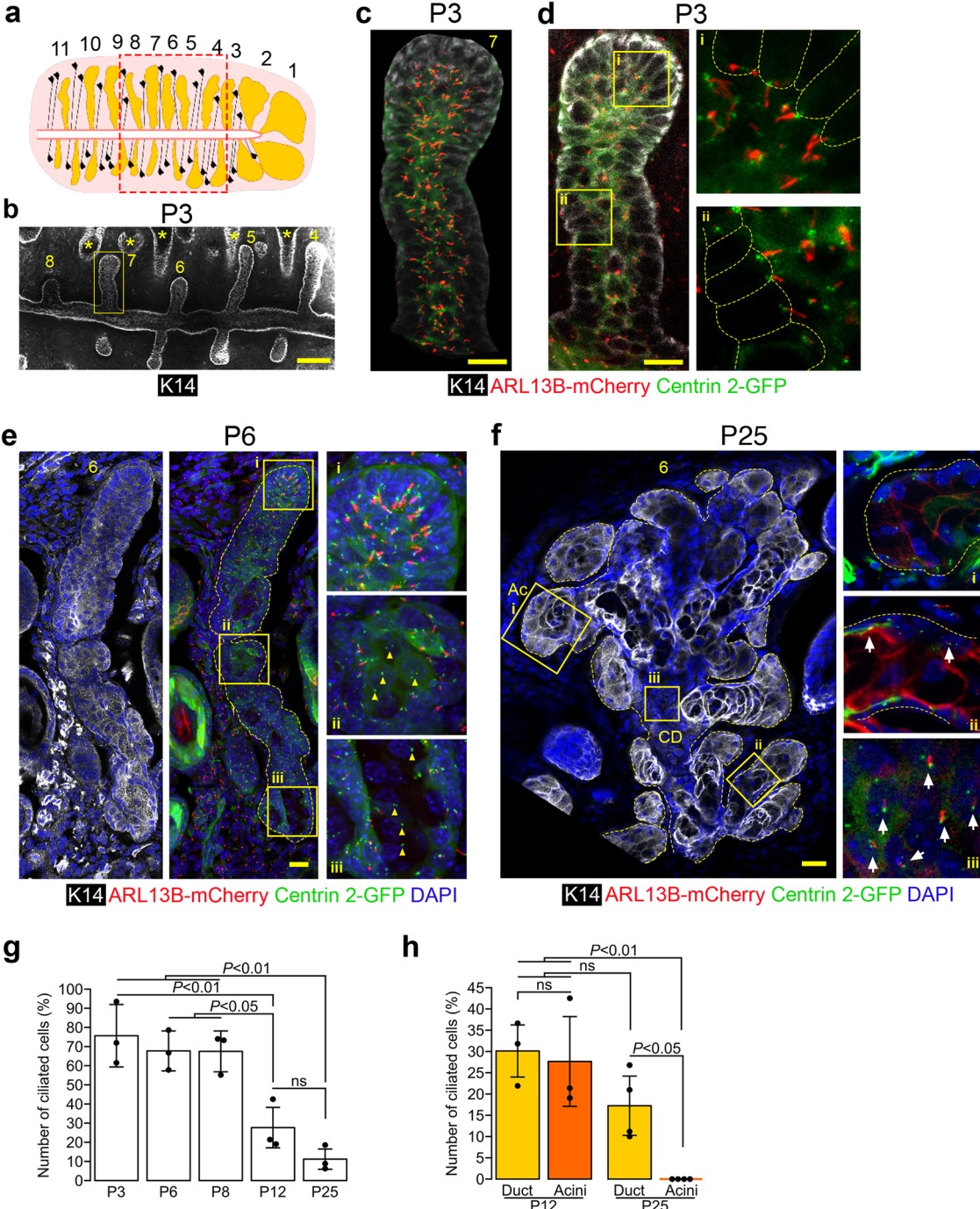

g, h: Number of ciliated cells (%)

K14 ARL13B-mCherry Centrin 2-GFP DAPI

not display cilia (Fig. 3e, yellow arrowheads). As MGs continued to mature, the percentage of ciliated cells progressively decreased to 30% and 12% at P12 and P25, respectively (Fig. 3g). The percentage of ciliated cells was similar in the duct and acini of MGs at P12 (Fig. 3h); however, at P25, primary cilia were absent in acini (Fig. 3f, h). At P25, most primary cilia were detected on the basal cells of the proximal region of the central duct. Still, some were also occasionally visible on the basal cells of the connecting ductules (Fig. 3f). Thus, primary cilia localized to basal cells of the developing MGs and disassembled as MG development progressed and meibocytes matured. However, cilia persisted on basal cells of the central duct of mature glands. Altogether, these results suggest a role of cilia during an early developmental step of MG morphogenesis primarily involving dividing and undifferentiated cells.

**Primary cilium ablation leads to the mislocalization of proliferating and apoptotic cells along the proximal-distal MG axis.** Several studies have shown that primary cilium resorption is required for cell proliferation and have demonstrated either reduction or loss of primary cilia in various epithelial neoplasms (reviewed in[39–41]). Thus, we examined whether the abnormal expansion in size of MGs of the cilia mutant was due to an increase in cell proliferation rates. Proliferation rates were determined by counting the number of EdU-positive cells 6 h after EdU injection and normalized to the total number of cells identified by DAPI nuclear staining. To accurately assess the gland proliferation rates, we counted EdU-positive and DAPI-positive cells on serial sections of the eyelid, covering the total MGs thickness for each gland. Cell proliferation rates were assessed at P4, P6, and P21 when MGs were elongating,

**Fig. 3 Most MG cells were ciliated in early stages of MG development, but meibocytes lost their cilium as they matured. a** To facilitate the comparison of similar MGs, MGs were numbered from the temporal side (MG#1) to the nasal side (MG#11). The boxed area indicates the region in which MGs were studied for all the following experiments **b** Whole mount tarsal plate at P3 imaged by confocal microscopy. MGs were stained with an antibody against K14 (in white). Hair follicles (marked with *) were also stained but were easily distinguishable from MGs due to the presence of a hair shaft. The boxed region indicates the MG shown at higher magnification in **c** and **d**. Scale bar, 100 μm. **c** 3D reconstruction with Imaris of MG#7 In Arl13b-mCherry;Centrin2-GFP mice, mCherry labels primary cilia in red, and GFP labels basal bodies in green. At P3, cilia were visible all along the MG. Scale bar, 15 μm. **d** Optical section picked in the center of MG#7. Basal bodies (in green) and primary cilia (in red) were localized on the apical side of MG basal cells (outlined with a yellow dotted line). Scale bar, 15 μm. **e** Representative MG longitudinal section at P6. MGs (outlined by a yellow dotted line) were stained with an antibody against K14 (in white), nuclei were stained with DAPI (in blue), basal bodies were labeled with GFP (in green) and primary cilia were labeled with mCherry (in red). MG cells are ciliated all along the MG, including in the distal tip of the gland (i), the forming acini (ii) and the forming central duct (iii). Scale bar, 15 μm. **f** Representative longitudinal section of a morphologically-mature MG at P25. MGs (outlined by a yellow dotted line) were stained with an antibody against K14 (in white), nuclei were stained with DAPI (in blue), basal bodies were labeled with GFP (in green) and primary cilia were labeled with mCherry (in red). No primary cilia were visible in the acini (i), but primary cilia (arrows) were still present in ductules (ii) and the central duct (iii). Scale bar, 100 μm; Ac, acini; CD, central duct. **g** Quantification of the percentage of ciliated cells throughout development ($n = 3$ for each age). Data were presented as mean ± SD. Statistical significance was assessed using Kruskal-Wallis test. For clarity, only statistically significant differences are indicated on the graph. **h** Spatial distribution of ciliated cells within MGs at P12 and P25. The percentage of ciliated cells was quantified specifically in the acini and in the central duct of MGs at P12 and P25 ($n = 3$ for each age). Data were presented as mean ± SD. Statistical significance was assessed using Mann Whitney test (Ctrl vs. cKO) and Wilcoxon signed rank test (acini vs. duct). ns, non-significant, $P \geq 0.05$.

beginning to branch, and reaching their mature morphology, respectively[10]. The overall percentage of dividing cells in MGs of both mutant and control mice was ~40% at P4, decreased to ~25% at P6, and reached baseline values of ~5% at P21 with no significant differences detected between control and cKO mice at any of the time points analyzed (Fig. 4a–d).

Furthermore, we examined the distribution of proliferating cells along the proximal-distal axis of the glands. In control mice, 52% and 28% of the cells were dividing in the distal half of the gland at P4 and P6, respectively. In contrast, only 32% and 22% of cells were proliferating in the proximal half of the gland in control mice at P4 and P6, respectively. Thus, during the development of control MGs, proliferating cells were more abundant in the distal region than proximal (Fig. 4a, b, e). In contrast, proliferating cells were distributed uniformly along the length of the glands in the mutant (Fig. 4a', b', e). By P21, the distal enrichment of proliferative cells along the proximo-distal axis was no longer visible (Fig. 4c, e). Thus, during MG development, primary cilia ablation did not alter the overall cell proliferation rates but, instead, had an impact on preferentially concentrating the dividing cells to the distal half of the glands. However, the number of basal bodies associated with a primary cilium was not significantly different between the distal and proximal halves of MGs at P3, P6, and P8 (Supplement Fig. 3).

Next, we assessed whether the perturbation of the gland architecture, observed with cilia ablation, similarly altered the number or distribution of apoptotic cells. We quantified the percentage of apoptotic cells in mature MGs at P21 by TUNEL and DAPI staining. The overall percentage of TUNEL positive (TUNEL + ) cells detected in the ductules, acini and the gland central duct, where the majority of apoptotic cells localized, was similar in both control and cKO mice (Fig. 4f, g). Although we didn't detect spatial segregation of apoptotic cells in the MG of both control and cKO mice (Fig. 4g), when we considered only the gland central duct, a higher percentage of TUNEL + cells were localized to the distal half than to the proximal half in control mice (Fig. 4h). In contrast, in the central duct of cKO mice, apoptotic cells were uniformly distributed between the distal and proximal half of the glands (Fig. 4h). Thus, the ablation of primary cilia does not affect rates of cell death, but instead affects the localization of TUNEL + cells within the MG central duct. Collectively, these results indicate that the absence of cilia affects the distribution of dividing and apoptotic cells rather than the overall rates of proliferation and cell death, which subsequently alters the architecture and size of mutant MGs.

**Primary cilia orchestrate cell patterning during early MG development, control the central duct width and overall MG size but not MG branching**. To determine how the absence of cilia affects MG morphogenesis and leads to abnormally larger glands, we examined the formation of cell patterns during early morphogenetic steps of MG development. The mG reporter expressed upon Cre-dependent homologous recombination in the mT/mG mouse allowed us to follow MG morphogenesis at the cell resolution in both mutant and control. At P1, when the epithelial invagination from the eyelid margin elongates into the eyelid mesenchyme, the meibomian anlages of the mutant displayed a similar size and overall shape of those observed in the control (Fig. 5a, b). However, while the cells of the invaginating epithelial anlage in control appeared well organized in one layer of basal cells surrounding a layer of suprabasal cells, in the mutant, this cellular pattern appeared less defined, and the basal and inner layers not clearly recognizable (Fig. 5a, b). As morphogenesis progressed, by P3 MG, primordia of the mutant were shorter but wider than those of the control (Fig. 5c–g). Moreover, the ratio of the number of basal cells compared to the number of cells in the center of the gland was significantly reduced in cKO mice, indicating that there were more cells in the central part of cKO MGs (Fig. 5h, i).

As MGs continue to develop, cells within the central cord of the epithelium expand at P6, the central duct begins to form, and lateral branches appear[10]. At P6 and P21, the width of the mutant central duct was 10% and 30% larger than the control, respectively (Fig. 6a–d). Because cell proliferation, apoptosis, and cell size remained unchanged in the MGs of both mutant and control, we reasoned that, at least during early developmental stages, the overall mass of the glands in both genotypes would remain similar. Using two-photon microscopy on whole mount samples of tarsal plates, we found no significant differences in the overall average volume between mutant and control MGs at P1, P3, or P4 (Fig. 6e, f). In contrast, at P8, the average volume of the mutant glands was nearly double that of the control glands (Fig. 6g, h). Finally, the number of acini was not significantly different between both genotypes (Fig. 6i). Thus, primary cilia promote the segregation of proliferating cells at the distal tip of the growing MGs, which in turn ensures a balanced growth in length and width of the MG duct (Fig. 6j). To determine whether ablation of cilia perturbs the outcome of known cilia-mediated signaling cascades, we examined the expression levels of selected genes known to be transcriptional targets or essential components of the Hedgehog (Hh), Wnt or Notch pathways in the tarsal plate

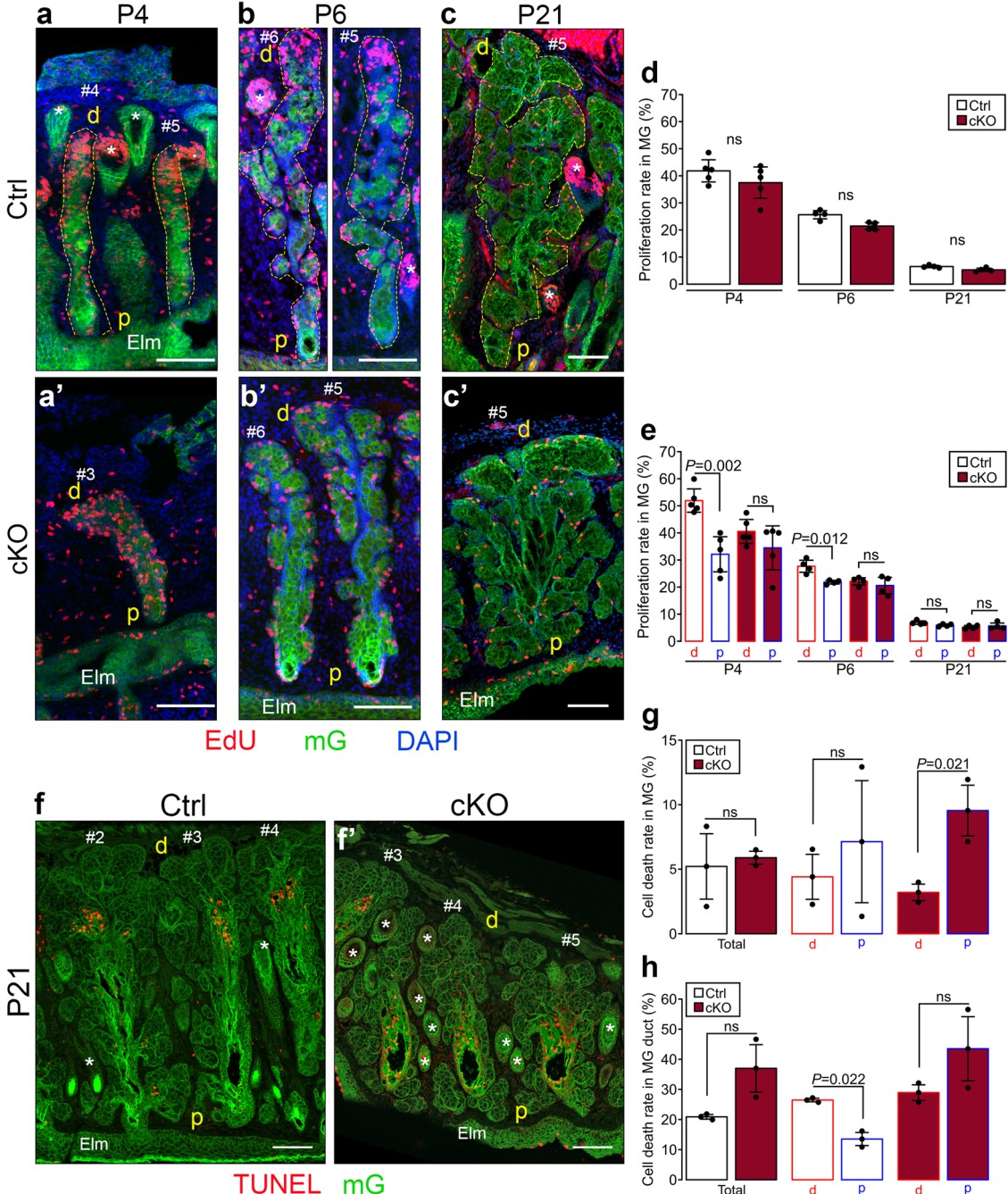

**Fig. 4 Primary cilium ablation did not change the overall rates of proliferation and dying cells in MGs but led to an abnormal distal/proximal localization of the proliferating and dying cells within MGs. a–c** Cell proliferation was assessed by EdU staining in MGs at P4, P6 and P21 in control (**a**, **b**, and **c**) and cKO (**a'**, **b'**, and **c'**) mice, respectively. Scale bar, 100 µm; *, hair follicle; Elm, eyelid margin; d, distal; p, proximal. **d–e** Proliferation rates were quantified at P4, P6 and P21 in the full MGs (**d**), and specifically in the proximal (p) half (from the eyelid margin to the center of the gland) and the distal (d) half (from the middle to the tip of the gland) of MGs **e**. Proliferation rates were determined by normalizing the number of EdU-positive nuclei to the total number of nuclei stained by DAPI (n = 4/group). Data were presented as mean ± SD. Statistical significance was assessed using the Mann Whitney test (Ctrl vs. cKO) and Wilcoxon signed rank test (distal vs. proximal). ns, non-significant, $P \geq 0.05$. (**f**) Cell death was assessed by TUNEL staining in MGs at P21 in control (**f**) and cKO (**f'**) mice, respectively. Scale bar, 100 µm; *, hair follicle; Elm, eyelid margin; d, distal; p, proximal. (**g–h** Cell death rates were quantified at P21 in the full MGs (**g**) and specifically in the central duct (**h**). Cell death rates were determined by normalizing the number of TUNEL-positive nuclei to the total number of nuclei stained by DAPI (n = 3/group). Data were presented as mean ± SD. Statistical significance was assessed using the Mann-Whitney test (total, Ctrl vs. cKO) and Wilcoxon signed rank test (distal vs. proximal). ns, non-significant, $P \geq 0.05$.

of adult cKO and control mice by quantitative real-time PCR (RT-qPCR). The expression levels of genes associated with the Wnt and Notch signaling cascades detected in the mutant were indistinguishable from those found in the control (Fig. 6k). In contrast, we detected a significant reduction of mRNA levels of

the *Gli1* gene, a transcription factor and a transcriptional target of the Hh pathway, in the mutant compared to the control (Fig. 6k). However, expression levels of additional genes of the Hh pathway, including the Hh ligands Desert (Dhh), Indian (Ihh) and Sonic hedgehog (Shh) and transcriptional targets, including

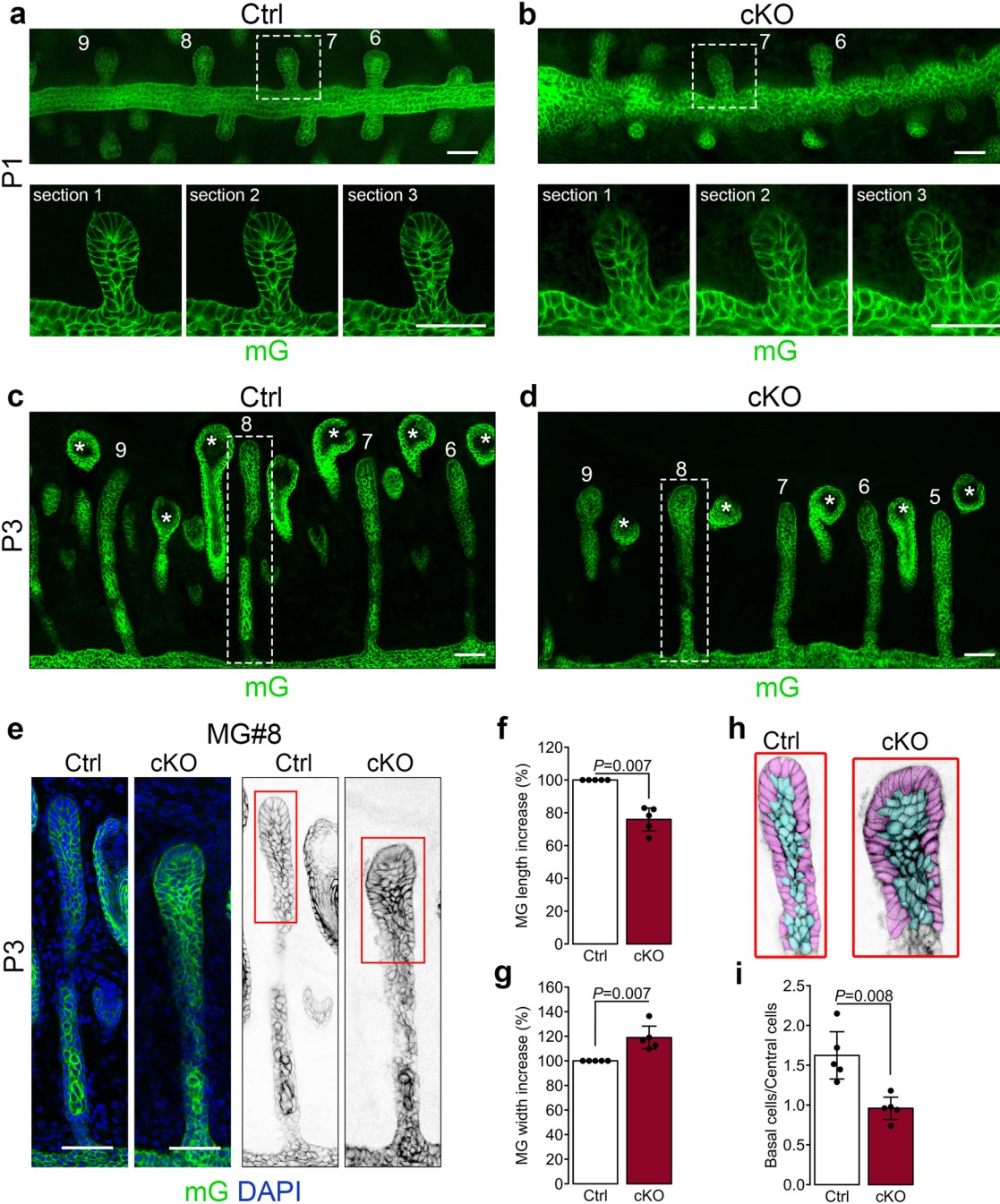

**Fig. 5 Primary cilium regulates MG elongation. a–b** Representative overview of whole mount tarsal plates at P1 imaged by confocal microscopy. MGs were visualized by the mG fluorescent reporter. Boxed regions indicate MGs shown at higher magnification, for which serial optical sections are displayed. Scale bar, 50 μm. **c–d** A representative overview of whole mount tarsal plates at P3 imaged by confocal microscopy. MGs were visualized by the mG fluorescent reporter. Boxed regions indicate MGs shown at higher magnification in **e**. Scale bar, 50 μm. MG length (**f**) and MG width (**g**) of cKO mice normalized to control ($n = 5$ mice/group). Data were presented as mean ± SD. Statistical significance was assessed using the Mann-Whitney test. **h** Basal cells (in purple) and central cells (in blue) were manually color-coded and then counted. **i** The ratio between basal cells and central cells was calculated ($n = 5$ mice/group). Data were presented as mean ± SD. Statistical significance was assessed using the Mann-Whitney test.

CyclinD and Ptch1, remained unchanged in both. Future studies involving single-cell analysis will further our understanding of the role of primary cilia in MG morphogenesis and renewal.

## Discussion

The primary function of the MGs is to secrete the meibum, a lipid layer that protects the ocular surface from hazardous environmental factors and desiccation[1,42,43]. Several factors can lead to functional defects of MGs for which there are no effective long-

term treatments. Despite their critical role in maintaining clear vision, the molecular mechanisms and signaling networks underlying the development and maintenance of MGs remain poorly understood. Our study unveils the critical role of primary cilia in controlling the size of MGs and the amount of meibum produced. These findings shed light on the fundamental mechanisms of MG development and maintenance with important implications for designing MGD treatments. Here we have shown that a mutant mouse lacking cilia, via ablation of the *Ift88* gene in K14-expressing

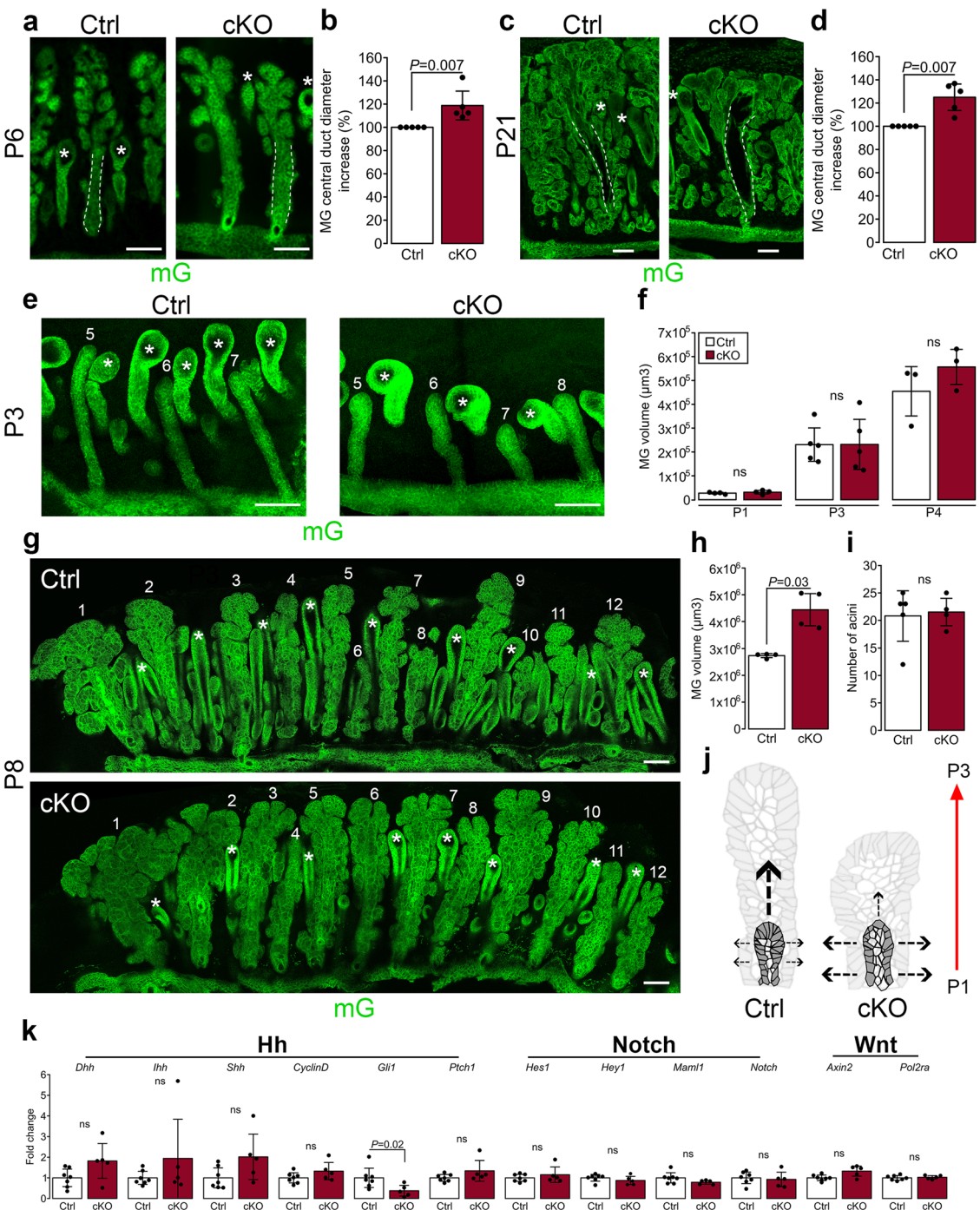

**Fig. 6 Primary cilium ablation induces dilation of the MG central duct and volume but does not affect MG branching. a–d** Representative MG longitudinal sections at P6 (**a**) and P21 (**c**) and quantification of central duct diameter at P6 (**b**) and P21 (**d**). MGs were visualized by the mG fluorescent reporter, and MG central duct diameter (outlined by white dotted lines) was measured ($n = 5$ mice/group). Scale bar, 100 μm. **e** Representative overview of whole mount tarsal plates at P3 imaged by 2-photon. Scale bar, 100 μm. **f** MG volume at P1, P3, and P4 was quantified after 3D reconstruction of z-stacks using Imaris ($n = 4$ mice/group at P1, $n = 5$ mice/group at P3 and $n = 3$ mice/group at P4). Data were presented as mean ± SD. Statistical significance was assessed using the Kruskal-Wallis test. ns, non-significant, $P \geq 0.05$. **g** Representative overview of whole mount tarsal plates at P8 imaged by 2-photon. Scale bar, 100 μm. **h** MG volume at P8 was quantified after 3D reconstruction of z-stacks using Imaris ($n = 4$ mice/group). Data were presented as mean ± SD. Statistical significance was assessed using the Kruskal-Wallis test. ns, non-significant, $P \geq 0.05$. **i** MG branching was assessed by counting the number of acini per gland at P8 ($n = 5$ controls and $n = 4$ cKO). Data were presented as mean ± SD. Statistical significance was assessed using the Mann-Whitney test. ns, non-significant, $P \geq 0.05$. (**j**) Model of MG morphogenesis and cell patterning formation occurring during early development (from P1 to P3) in cKO and control. **k** RT-qPCR analysis of Hh (*Dhh, Ihh, Shh, CyclinD, Gli1, Ptch1*), Notch (*Hes1, Hey1, Maml1, Notch1*), Wnt (*Axin2*) target genes in isolated tarsal plates of cKO and control shows significant downregulation of *Gli1* expression in mutant tissue relative to control at adulthood ($n = 7$ controls and $n = 5$ cKO). Data were presented as mean ± SD. Statistical significance was assessed using the Mann-Whitney test. ns, non-significant, $P \geq 0.05$.

tissue, develops abnormally large MGs, which contain twice as much lipids as usual. However, gland enlargement was not achieved by alterations of proliferation or apoptosis rates. Instead, we demonstrated that the ablation of cilia in developing glands altered the cellular organization and the localization of dividing cells along their proximal-distal axis, modifying their cellular patterning and resulting in increased gland dimensions.

Using a transgenic mouse model with fluorescently labeled cilia and basal body, we have shown that primary cilia were present on meibocytes primarily during early MG development. Thus, to explain how developmental processes in cilia-deficient K14-expressing-tissue led to larger MGs in adult mice without changes in proliferation or apoptosis rates, we have examined morphogenesis and cell patterning of the developing MGs in control and cKO mice. We found that at P1, the cellular pattern of control glands budding from the fused eyelid margin appeared organized in one bordering layer of basal cells and one layer of suprabasal cells at the center of the elongating bud. In contrast, this cell pattern was perturbed in the MG buds of the cilia mutant, where cells appeared randomly organized with poorly discernible layers (Fig. 5). At P3, the solid epithelial cord continued to invaginate into the mesenchyme of the tarsal plate, and the number of central/suprabasal layers increased to 2 or 3. During the elongation phase between P4 and P6, the majority of the dividing cells localized at the distal tip of the developing glands of the control but were uniformly distributed along the length of the glands in the mutant. Thus, cilia of the MGs could be essential in sensing mitogenic signals required for proximo-distal elongation. We propose that failure to spatially segregate dividing cells in the growing gland could interfere with the elongation process at the distal tip of the glands. Because no difference in proliferation was detected, slower growth at the distal tip would consequently lead to an abnormal lateral expansion of the invaginating epithelial cords. Indeed, we found that the number of central/suprabasal cell layers abnormally increased to 4 or 5 in the gland of the cilia mutant, and as a result, developing glands were both wider and shorter than the control glands. Consistent with this possibility, the total volume of the glands remained similar in both mutant and control up to at least P4 (see the model in Fig. 6j). At P8, the volume of the mutant MGs increased significantly compared to the control; however, the gland length was similar in both. Thus, we hypothesize that as the glands grow, inhibitory signals could hinder their elongation, which eventually halts at full morphological maturation of the glands (P15). Therefore, while control MGs stopped elongating, mutant glands continued to grow and eventually reached the length of control glands. However, because the central duct was larger in the mutant glands, the overall size and lipid content of the mutant MGs were increased relative to the control. Similar dynamics could also occur during the genesis of the ductules. Future studies to elucidate the role of primary cilia in determining the localization of dividing cells at the distal tip of the developing MG may reveal a novel role for cilia in tissue morphogenesis.

Unlike SGs, MGs are not structurally linked to the hair follicle[1]. However, it is unclear whether the hair follicles of the eyelashes, which intercalate the MGs affect their development or homeostasis. In the skin, ablation of ciliogenic proteins in K14-expressing tissue leads to degeneration of the hair follicles via inactivation of the Hh signaling pathway[23,29,44]. However, defects in hair follicle formation and maintenance were evident only three weeks after birth, a point at which the MGs have already reached their mature configuration[23]. Thus, we can exclude any indirect role of the hair follicles on cilia-dependent defects affecting the MGs as reported in this study. It is widely accepted that the ciliary compartment is required for the propagation of the Hh signaling pathway and that ablation of Ift88 inhibits Hh

responsiveness in several tissues[45,46]. Accordingly, we detected reduced mRNA levels of the transcription factor and transcriptional target gene Gli1 in the mutant tarsal plate when compared to the control. Studies in the skin have shown that the Hh pathway is critical for the development of SG, which share several basic features with the MGs[47]. Thus, our results showing abnormal enlargement of cilia-deficient MGs via Ift88 ablation may be inconsistent with Hh mediating MG development. Interestingly, however, ablation of cilia in K14-expressing cells compromised hair growth and maintenance via depletion of the Hh signaling but also resulted in enlarged and multilobulated SG in the mouse tail[23]. Thus, this evidence suggested an unusual and complex involvement of cilia in the cilia-Hh signaling axis in SGs and MGs. A thorough investigation of this interaction could reveal fundamental mechanistic insights in designing therapeutic strategies to address both skin-related conditions associated with ciliopathies, such as keratosis pilaris and seborrheic dermatitis, and MGD[32].

Here we have shown that meibocytes lose their primary cilium as they mature. Similar to other epithelia, including the epidermis and the corneal epithelium, cilia were present on basal cells but disassembled as cells differentiated and moved apically[22–24]. In mature glands, ciliated cells were restricted to the proximal region of the central duct and at the basal layer of the ductules and acini. Intriguingly, it was demonstrated that in adult MGs, slow-dividing cells, which reflect a stem cell characteristic, localize to the ductules at the point where the central duct transitions to the acini[48]. Thus, it would be interesting to determine whether slow-cycling cells are ciliated. In the skin, primary cilia play a Hh-independent interfollicular role in the stratification of the epidermis at homeostasis[23,24]. Ablation of Ift88 increased rates of proliferation and led to an expansion of the basal layer with basal-like cells. However, hyperproliferation did not lead to the formation of tumors or blistering[23]. This is in contrast with our finding indicating that the expansion of MG does not involve an increase in proliferation rate. However, it must be noted that ablation of Ift88 in highly proliferating corneal epithelium during development or repair had no effect on proliferation rates[22]. Suggesting that cell proliferation could be a secondary and indirect effect of cilia ablation. Interestingly, in the skin and cornea primary cilia of epithelial basal cells modulated the Notch pathway independently of cell proliferation[22,24]. Although our results from RT-qPCR of tarsal plate tissue suggest that the primary cilium in MGs of adult mice are not required to transduce the Notch signaling, it is possible that potential differences in mRNA levels of Notch target genes between mutant and control are under the detectable margin of this approach. Indeed, we have shown that only a small number of cells remain ciliated in mature MGs. Thus, future studies at the single-cell level could provide a more comprehensive understanding of cilia involvement in Notch and MG maintenance. Interestingly, it was shown that suppression of the Notch signaling pathway in progenitor cells of SGs led to atrophy of the gland[49]. In contrast, Notch ablation outside the stem cell compartment drove SG expansion[49]. In another study, ablation of Notch1 in K14-expressing tissue led to the formation of cyst-like structures replacing the MGs[50]. Thus, it would be interesting to determine a possible involvement of the primary cilium in the transduction of the Notch pathway at the single cell level in relation to stem cell niche maintenance in MGs[51].

Lipid analysis has shown that ablation of cilia during MG development led to a two-fold increase in lipid content. Although we have demonstrated the involvement of cilia in regulating MG size, it remains unclear whether the cilium also plays a more direct role in meibum production. The Peroxisome proliferator-activated receptor-γ (PPARγ) is a member of the nuclear receptor family of ligand-activated transcription factors implicated in

regulating adipocyte and sebocyte differentiation as well as lipogenesis[52,53]. Moreover, PPARγ is also involved in meibocyte differentiation in vivo and in vitro and is required for the upregulation of genes implicated in lipid production[54–56]. Recent studies have shown that in the context of injury-induced adipogenesis, ablation of cilia in fibro/adipogenic progenitors (FAP) via *Ift88* deletion inhibited the production of PPARγ and FAP differentiation into adipocytes[57,58]. These studies would argue against a direct involvement of the MG cilia in PPARγ-dependent lipogenesis or meibocyte differentiation since cilia ablation in MGs led to an increase in lipid amount and an expansion of meibocyte mass. However, unlike most cell types, primary cilia of FAPs and cells in the developing limb bud inhibit GLI1 and PATCHED1 expression by promoting GLI3 repressor formation. Consequently, the ablation of cilia led to an increase, rather than a decrease, of the Hh signaling activity[59]. Several studies have shown the complex role of the Hh pathway in adipogenesis and PPARγ regulation[60–63]. Thus, future investigation addressing the role of GLI proteins and, more broadly, the role of the Hh pathway in MGs development and homeostasis will provide critical mechanistic insights into our understanding of meibocyte differentiation, renewal, and meibogenesis.

To date, treatment options for MGD and DED are limited. Physical treatments seeking to increase the quality and quantity of meibum did not achieve satisfactory long-term results[64–66]. Other approaches aiming to substitute the lipid layer with topical application of lipid-containing artificial tears and emulsions are challenging given the complex structure and composition of the lipid layer of the ocular surface[42,67] Thus, expanding treatment strategies by directly targeting MG renewal and lipid production have the potential to not only relieve ocular surface discomfort but also improve the quality of life in affected patients. This work revealed that cilia-mediated pathways control MGs expansion and lipid production without affecting the lipid composition, pointing to a novel therapeutic target to combat MGD.

## Methods

**Mice**. Mouse strains *Ift88^{tm1Bky}* (here referred to as *Ift88^{fl/fl}*)[68], B6N.Cg-Tg(KRT14-cre)1Amc/J (*K14-Cre*, Jackson Laboratory stock No 018964)[69], and Gt(Rosa)26Sor(tm4(ACTB-tdTomato,-EGFP)Luo)/J (*mT/mG*), Jackson Laboratory stock No 007676)[34] were maintained on mixed C57Bl/6, FVB and 129 genetic backgrounds. The *Ift88* conditional knockouts (cKO) were generated by crossing *K14-Cre;Ift88^{fl/+}* males with *Ift88^{fl/fl}* females. Other allelic combinations than *K14-Cre;Ift88^{fl/fl}* (cKO) were considered as controls (Ctrl). Mouse strain Tg(CAG-Arl13b/mCherry)1 K and Tg(CAG-EGFP/CETN2)3-4Jgg/KandJ (here referred to as Arl13b-mCherry;Centrin2-GFP)[37] was purchased at Jackson Laboratory (stock No 027967). All animal procedures were performed in accordance with the guidelines and approval of the Animal Care and Use Committee at Johns Hopkins University and with the ARVO Statement for the Use of Animals on Ophthalmic and Vision Research.

**Histology and immunofluorescence staining**. Upper and lower eyelids from P6 and P21 mice were dissected, fixed overnight in 4% paraformaldehyde (PFA) in PBS, and embedded in paraffin for histological analysis. Hematoxylin and eosin (HE) staining was performed following standard procedures. Sections were imaged with an Olympus slide scanner VS200 (Olympus, Center Valley, PA). Eyelids from P3 mice were dissected, fixed for 30 min to 2 h in 4% PFA in PBS and embedded in optimal cutting temperature compound (OCT Tissue-Tek, Sakura Finetek, Torrance, CA). Cryosections were processed for primary cilium staining. After 10 min fixation with cold acetone (−20 °C) and 20 min permeabilization with 0.5% Triton X-100 in PBS, sections were incubated with a mouse anti-acetylated tubulin antibody (1:1000, T6793, Sigma-Aldrich, St Louis, MO) and/or a rabbit anti-ARL13B antibody (1:800, 17711-1-AP, ProteinTech Group, Rosemont, IL) in 2% BSA/0.1% Triton X-100/PBS overnight at 4 °C. Sections were then incubated in secondary fluorescent antibodies donkey anti-rabbit Alexa Fluor™ 647 (1:500, A-31573, Thermo Fisher Scientific, Waltham, MA), donkey anti-mouse fluorescein (FITC) (1:200, 715-095-150, Jackson ImmunoResearch Laboratories Inc., West Grove, PA) and donkey anti-mouse Rhodamine (TRITC) (1:200, 715-025-150, Jackson ImmunoResearch Laboratories Inc.) in 2% BSA/PBS for 2 h. Sections were mounted with VECTASHIELD Antifade Mounting Medium (H-1000, Burlingame, CA) and imaged with a Zeiss LSM880 confocal microscope (Zeiss, Jena, Germany).

**Fluorescent imaging of whole mount MGs**. Eyelids from P1, P3, P4 and P8 *K14-Cre;Ift88^{fl/fl};mT/mG* mice (cKO) and *K14-Cre;Ift88^{fl/+};mT/mG* (Ctrl) littermates were dissected, fixed in 4% PFA in PBS for 1 h and washed in PBS. During the fixation, most of the connective tissues and muscles covering the tarsal plate were manually removed. MGs were mounted in 90% glycerol and imaged with a LSM880 confocal microscope (for samples at P1) or a LSM710/NLO two-photon microscope (for samples at P3, P4 and P8). Serial optical sections were acquired in 1 or 2 μm steps through the entire MG. MG volume was quantified after 3D reconstruction of the z-stacks with Imaris (Bitplane, South Windsor, CT), and the number of acini per MG was manually counted.

**Oil Red O (ORO) staining in whole-mount MGs**. Eyelids from P6, P8, and P21 mice were dissected, fixed in 4% PFA in PBS for 1 h and washed in PBS. During the fixation, most of the connective tissues and muscles covering the tarsal plate were removed. MGs were stained for 1 h in ORO solution (Electron Microscopy Sciences, Hatfield, PA) at room temperature (RT), rinsed with distilled H₂O, mounted in 90% glycerol and imaged with an Olympus MVX10 dissecting scope (Olympus). MG size was determined by averaging the MG area of individual MGs measured with Fiji[70].

**Cilia localization**. Eyelids from P3, P6, P8, P12 and P25 Arl13b-mCherry;Centrin2-GFP mice were dissected and fixed for 1 h in 4% PFA/1% Triton X-100 (Mallinckrodt Pharmaceuticals, Staines-upon-Thames, United Kingdom) in PBS. Eyelids were then processed for K14 staining on whole MGs or embedded in OCT. For whole MGs, prior to staining, most of the connective tissues and muscles covering the tarsal plate were removed and tarsal plates were permeabilized for 1 h with 2% BSA/1% Triton X-100/PBS at RT. MGs for whole mount samples and cryosections were stained with a rabbit polyclonal antibody directed to K14 (1:1000, 905301, BioLegend, San Diego, CA). Nuclei were counterstained with DAPI on cryosections. MG whole mounts (P3, P6, and P8) and cryosections (P12 and P25) were imaged with a Zeiss LSM880 confocal microscope. Serial optical sections were acquired in 1 μm steps. After 3D reconstruction of the z-stacks with Imaris, MGs were outlined using the Surfaces tool, and primary cilia and basal bodies were counted in MGs using the Spots tool. Centrin2-GFP-labeled centrioles were considered as a single basal body when less than 2 μm apart. Since the number of basal bodies was similar to the number of nuclei as shown in Supplement Fig. 4, the number of ciliated cells in MGs was determined by normalizing the number of primary cilia to the number of basal bodies and expressed as a percentage.

**In vivo cell proliferation assay**. Mice at P4, P6 and P21 received a single intraperitoneal injection of 50 mg/kg EdU (EdU-Click 594, baseclick, Germany) and were sacrificed after 6 h, as described elsewhere[71]. Eyelids were dissected and snap-frozen in OCT. Twenty-micron cryosections were processed following manufacturer instructions (EdU-Click 594, baseclick, Germany). Briefly, sections were fixed for 15 min with 4% PFA in PBS, permeabilized for 20 min with 0.5% Triton X-100 in PBS and then stained for 30 min with the reaction cocktail in the dark at RT. MGs were localized using the mT/mG fluorescent reporter or stained using a rabbit anti-K14 polyclonal antibody (1:1000, 905301, BioLegend). Nuclei were counterstained with DAPI. Sections were imaged with a Leica DMI6000 microscope equipped with a Yokogawa confocal spinning disc or a Zeiss LSM880 confocal microscope. For each cryosection, serial optical sections were acquired in 2.45 μm steps. For each mouse, serial cryosections were processed to acquire the entirety of the MGs. After 3D reconstruction of the z-stacks with Imaris, MGs were outlined using the Surfaces tool, and EdU-positive nuclei and DAPI-positive nuclei were counted in each MG using the Spots tool. Quantification was performed on the MGs located in the center of the upper eyelid. The cell proliferation rate per MG was determined by normalizing the number of EdU-positive nuclei to the number of DAPI-positive nuclei. Quantification of the EdU-positive and DAPI-positive nuclei was also performed, separating the proximal part (from the eyelid margin to the middle of the gland) and the distal part (from the middle of the gland to the tip) of the MGs.

**Cell death TUNEL assay**. Eyelids from P21 mice were dissected, fixed overnight with 4% PFA in PBS and embedded in paraffin. Sections were processed for apoptosis immunofluorescent staining using the In situ Cell Death Detection Kit, TMR Red (Roche Applied Science, Mannheim, Germany), as described in[72]. Sections were permeabilized for 15 min with 10 μg/mL proteinase K in 10 mmol/L Tris/HCl (pH 7.4) at RT and then stained for 1 h with the reaction mixture at 37 °C in the dark. Nuclei were counterstained with DAPI. Sections were imaged with a Zeiss LSM880 confocal microscope. TUNEL-positive and DAPI-positive nuclei were counted on 3 serial sections, 20 μm apart from each other. The cell death rate per MG was determined by normalizing the number of TUNEL-positive nuclei to the number of DAPI-positive nuclei.

**Quantitative RT-qPCR**. Tarsal plates were isolated by dissection and were immediately submerged in RNAlater (AM7020, ThermoFisher, Waltham, MA) and stored at −20 °C for up to 1 month. RNA was extracted from eyelids using the RNeasy mini kit (Qiagen, Germantown, MD) as per the manufacturer's

**Table 1 RT-qPCR primers.**

| Gene | Forward | Reverse | Reference |
|---|---|---|---|
| *Axin2* | 5′-CTCCCCACCTTGAATGAAGA-3′ | 5′-ACTGGGTCGCTTCTCTTGAA-3′ | 22 |
| *CyclinD* | 5′-AAGTGCGTGCAGAAGGAGAT | 5′-TTAGAGGCCACGAACATGC-3′ | 22 |
| *Dhh* | 5′-TGGCATTGTGAGTTTCCTCCT-3′ | 5′-AGCATGGACTTGGTTGGCTT-3′ | 79 |
| *Gapdh* | 5′-CATCACTGCCACCCAGAAGACTG-3′ | 5′-ATGCCAGTGAGCTTCCCGTTCAG-3′ | 22 |
| *Gli1* | 5′-TCCGGGCGGTTCCTACG-3′ | 5′-ACCATCCCAGCGGCAGTCT-3′ | 22 |
| *Hes1* | 5′-GGAAATGACTGTGAAGCACCTCC-3′ | 5′-GAAGCGGGTCACCTCGTTCATG-3′ | 22 |
| *Hey1* | 5′-CCAACGACATCGTCCCAGGTTT-3′ | 5′-CTGCTTCTCAAAGGCACTGGGT-3′ | 22 |
| *Ihh* | 5′-CTGCAAGGACCGTCTGAACT-3′ | 5′-TGGCTTTACAGCTGACAGGG-3′ | 22 |
| *Maml1* | 5′-CCAGCTTTGATGGCATATCTTCC-3′ | 5′-CTACAGGGACACTGGAAGGGTT-3′ | 22 |
| *Notch1* | 5′-GCTGCCTCTTTGATGGCTTCGA-3′ | 5′-CACATTCGGCACTGTTACAGCC-3′ | 22 |
| *Polr2a* | 5′-CGAGAAGGTCTCATTGACACGG-3′ | 5′-ACCACCTGGTTGATGGAGTTCC-3′ | MP211208, Origene, Rockville, MD |
| *Ptch1* | 5′-ACGGGGTCCTCGCTTACAAAC-3′ | 5′-TCTCGTAGGCCGTTGAGGTAGAA-3′ | 22 |
| *Shh* | 5′-TGTGTTCCGTTACCAGCGAC-3′ | 5′-AGCGAGGAAGCAAGGATCAC-3′ | 79 |

instructions. One microgram of RNA was reverse transcribed using the Super-Script III reverse transcriptase kit (18080051, ThermoFisher) as per the manufacturer's instructions. Quantitative PCR was carried out using SYBR green PCR master mix (4309155, ThermoFisher) in 20 μL in duplicate on a CFX96 qPCR (BioRad, Hercules, CA), machine using a 2-step cycle with annealing temperature of 60 °C. Quantification was carried out using the $2\text{-}\Delta\Delta^{cT}$ method[73] with the geometric mean of *Gapdh* and *Polr2a* used as normalization[74]. Primer sequences are listed in Table 1.

**Lipid analysis.** Meibomian lipids were extracted from surgically excised mouse tarsal plates (4 from each mouse) at 4 °C using three sequential extractions with a chloroform:methanol (3:1, vol:vol) solvent mixture. The extracts (3 x 1 mL) were pooled, and the solvent was evaporated under a stream of compressed nitrogen at 37 °C. The oily residue was redissolved in 1 mL of LC/MS quality iso-propanol and stored in a nitrogen-flushed, crimper-sealed HPLC 2-mL autoinjector vial at −80 °C before the analyses.

The gradient and isocratic reverse-phase liquid chromatography/high-resolution time-of-flight atmospheric pressure chemical ionization mass spectrometry (LC/MS) analyses were conducted using, correspondingly, an Acquity UPLC C18 (1 mm × 100 mm; 1.7 μm particle size) and an Acquity UPLC C8 (2 mm × 100 mm; 1.7 μm particle size) columns (both from Waters Corp., Milford, MA, USA) as described in detail in our earlier publications for mouse and human meibum[75–77]. Between 0.5 and 1.0 μL of the sample solution was injected per experiment. A Waters Acquity M-Class binary ultra-high performance LC system (UPLC, Waters Corp.) was operated at a 20 μL/min flow rate. The analytes were eluted using acetonitrile/iso-propanol isocratic solvent mixtures with 5% of 10 mM ammonium formate as an additive. The analytes were detected using a high-resolution Synapt G2-Si QToF mass spectrometer [equipped with a ZSpray interface, an IonSabre-II atmospheric pressure chemical ionization (APCI) ion source, and a LockSpray unit (all from Waters Corp.)]. All the experiments were conducted in the positive ion mode. Most of the lipids were detected as $(M + H)^+$ and $(M + H - H_2O)^+$ adducts. The major lipid analytes were identified using the EleComp routine of the MassLynx v.4.1 software package (Waters Corp.). However, some of the compounds, for example, triacylglycerols and cholesteryl esters, underwent spontaneous in-source fragmentation producing $(M + H - \text{fatty acid})^+$ species and were additionally characterized as $(M + Na)^+$, $(M + K)^+$, and $(M + NH_4)^+$ adducts. Finally, chromatographic retention times and mass spectra of major analytes were compared with those of authentic lipid standards (where available). The results of a detailed analysis of the MG lipidome of mutant mice are to be reported separately.

The total lipid production by MGs was estimated on the basis of the total ion chromatograms of their lipid extracts recorded in isocratic LC/MS APCI experiments as described recently[77]. The unbiased, untargeted analysis of the lipidomic data was conducted using Progenesis QI and EZinfo software packages (Waters Corp.). The Principal Component Analysis (PCA) approach was used to evaluate the differences between the control and cKO samples.

**Statistics and Reproducibility.** For each experiment, control and cKO mutant littermates from at least 2 different litters were compared. Data were presented as mean ± SD. Mann Whitney test was used to compare the cKO mutant to the control mice. Wilcoxon signed-rank test was used to compare different regions of the same glands (acini vs. duct and proximal vs. distal halves of MGs or central ducts), and the Kruskal-Wallis test with Dunn post-hoc test was used to compare the percentage of ciliated cells per gland throughout MG development and the MG volume throughout MG development. Statistical tests were performed with the online web statistical calculators https://astatsa.com/ or RStudio[78]. A *P* value < 0.05 was considered significant.

**Reporting summary**. Further information on research design is available in the Nature Portfolio Reporting Summary linked to this article.

## Data availability

Source data for the main and supplementary figures are provided as Supplementary Data 1. Extra data that support the findings of this study are available from the corresponding author (C.I.) upon reasonable request.

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

## Acknowledgements

The authors thank Qing Liu for his involvement during the early stages of this project, Hoku West-Foyle from the Microscope Facility at Johns Hopkins for his technical assistance, the reference histology core facility at Johns Hopkins University for the paraffin sections, and the members of the Wilmer Cornea Group for their helpful input and discussions. This work was supported by grants from the National Eye Institute, National Institute of Health (EY030661 to CI, EY024324 and EY027349 to IB), a core grant to the Wilmer Eye Institute (EY001765), a grant from the Office of Director, NIH (S10RR024550 to S.C. Kuo, Microscopy Facility at Johns Hopkins University); by a grant from the Eisinger Family; by a Wilmer Eye Institute Seed Fund to CI; and by an unrestricted grant from RPB to the Wilmer Eye Institute.

## Author contributions

C.Portal., I.B. and C.I. conceived and designed the study; C.Portal, Y.L., V.R., J.F., A.W., I.B. and C.I. performed experiments and analyzed and visualized data; C.Peterson and S.Y. provided conceptual and experimental guidance; I.B. and C.I. acquired funding; C.Portal, I.B. and C.I. wrote the paper with edits and feedback from all the authors. C.I. supervised and administered the project.

## Competing interests

The authors declare no competing interests.
