## [Peer Review File · Communications Biology]

Reviewers' comments:

Reviewer #1 (Remarks to the Author):

In "Primary cilia control cellular patterning..", Portal et al. investigated the role of cilia during meibomian gland development using targeted cilia-deficient mice. Dry eye disease caused by meibomian gland dysfunction (MGD) is very common degenerative condition and it is an important disease. This work provides the basis for a critical role for primary cilia in the patterning and formation of Meibomian glands. The authors provide a novel animal model to study MG in keratin-14 expressing tissue, showing that the loss of cilia results in moderate changes in MG.

The authors showed a critical role of primary cilia in controlling the size of MGs and the amount of meibum produced. They showed that the loss of cilia in developing MG changed the cellular organization and localization of dividing cells. The hypertrophy of MG increased the meibum volume but did not affect the lipid composition of MG. The loss of cilia in MGs development caused a significant increase in lipid content but without affecting the number of proliferating cells. Overall, the experiments are well performed and analyzed and it is an novel animal model for meibomian gland disease.

There are some minor criticisms.

The data shows that the ablation of cilia leads to an abnormal increase of lipid amount, which is not consistent with the most common mechanism for MGD, with a low delivery state characterized by gland obstruction. The underlying pathophysiology proposed to be terminal duct obstruction with decreased glandular secretion. The subtle change in meibum content is surprising.

Dry eyes related to MGD may present with corneal epithelial defects and erosions. Some corneal surface evaluations/images would be helpful.

A number of MGD treatments have been used clinically (doxycycline, cyclosporine, etc) does any of these reverse the phenotype observed?

Minor

3. There is no "c" in Fig. 3 (duplicate e) and no P25 in Fig. 3h at line 166 and 173 in the manuscript, respectively. Please confirm the consistency of the manuscript and figures.

Figure 6e should be log-scale or be two separate graphs (break in Y-axis seem to over-exaggerate the difference)

Reviewer #2 (Remarks to the Author):

The manuscript by Portal et al. unveils a new role for primary cilia in Meibomian glands (MGs). The authors found that primary cilia regulate MG size and lipid content, but not lipid composition, possibly by regulating cellular patterning during early MG development. The study is interesting and in general carefully performed, with adequate controls and quantification of the results. However, the authors do not go deep enough into the mechanisms that can explain the observed phenotypes. In particular, they do not investigate any ciliary-dependent signaling pathways that can be involved in the regulation of the cellular patterning differences detected (and some of them are referred in the discussion). Therefore, the authors should investigate the expression of Hh (and/or Notch) signaling components in cKO MGs. But potentially more interesting would be to investigate planar cell polarity (PCP). In kidney tubules, PCP dysregulation caused by abnormal primary cilia leads to enlarged ducts (and the formation of fluid-filled cysts that lead to cystic kidneys), which is somehow reminiscent of

the phenotype that the authors describe. Thus, it would be valuable to investigate the expression and localization of PCP components.

Other issues:

- Arl13b is used as a ciliary marker and, indeed, it localizes to the ciliary membrane. However, another marker should be used to ascertain cilia abrogation, namely acetylated tubulin, which is a marker of the axoneme core (at least by immunofluorescence), to validate the results of Suppl. Fig. 1.
- In Fig. 3, the authors quantify the % of basal bodies associated with a primary cilium, which is not a standard measurement. Even though each cell should have only one basal body, the results should be displayed as % of ciliated cells. Since the authors have nuclear staining, this should be straightforward.

Minor issue:

- In Fig. 6g, authors should label which schematic corresponds to control and cKO.

Reviewer #3 (Remarks to the Author):

Portal et al, address the question of the role of primary cilia in MG development and function. To do so, they use convincing mouse genetic models. Based on these, they claim that primary cilia ablation affects the segregation of dividing cells compartments along the proximal-distal MG axis resulting in larger MG with increased lipid production.

Although we find the study globally well-conducted and of interest to the biologist community, and of potential relevance for DED, there are major points we would like to mention:

Major points:

-Globally, quantifications are based on a low number of samples per condition. In Fig. 1d/ 2e/3g-h/4d-e-g-h/5f-g/6a-b-e-f, the authors should increase the number of samples to clarify the statistical significance.

- The choice of statistical tests (Mann-Whitney, Anova, Wilcoxon,...) is not clear. Please explain in the statistics section the choices for using each test. The statistical test used should be mentioned in every figure legend.

For example, in Fig. 3h/4a-e-g-h/5i/6e-f/S2a-b/S3a: Student t-test should be used when the variances are similar and Mann-Whitney test when the variances are different. Use Fisher test (F test) to compare the variances of the two groups. Anova should be systematically used for multiple groups comparisons. In Fig. 5f-g/6a-b, use one sample t comparing with 100.

Minor points:

- The order of the figures and lettering. The authors need to revise these, especially for Figure 4, as it makes the manuscript difficult to read.

- Fig. 2e: legend is unclear. Missing statistics.

- Figure 3: Fig 3b not mentioned in the text. In the figure "e" instead of "c" and no scale bars in 3c and 3d. 3h legend is unclear.

- Fig. 4d-e: please add the name of the measured areas on the graphs.

Point-by-point response

Dear Reviewers,

We sincerely appreciate your constructive criticisms and comments. Your suggestions, feedback, and remarks have led to additional experiments that have significantly strengthened the manuscript. We have substantially increased the number of biological samples throughout the study, improved statistical analyses, further ascertained the ablation of cilia in the conditional mutant mouse model using alternative cilia markers, assessed the expression of target genes of cilia-associated signaling cascades, and addressed issues relative to figure numbering and labeling. Based on your suggestions, we have revised our manuscript and used red color to indicate additions or modifications to the original text. Below, we discuss how we have attended to your comments point-by-point. We hope that our revised manuscript has addressed the reviewers' concerns and is now suitable for publication in *Communications Biology*.

Reviewer #1 (Remarks to the Author):

In “Primary cilia control cellular patterning..”, Portal et al. investigated the role of cilia during meibomian gland development using targeted cilia-deficient mice. Dry eye disease caused by meibomian gland dysfunction(MGD) is very common degenerative condition and it is an important disease. This work provides the basis for a critical role for primary cilia in the patterning and formation of Meibomian glands. The authors provide a novel animal model to study MG in keratin-14 expressing tissue, showing that the loss of cilia results in moderate changes in MG. The authors showed a critical role of primary cilia in controlling the size of MGs and the amount of meibum produced. They showed that the loss of cilia in developing MG changed the cellular organization and localization of dividing cells. The hypertrophy of MG increased the meibum volume but did not affect the lipid composition of MG. The loss of cilia in MGs development caused a significant increase in lipid content but without affecting the number of proliferating cells. Overall, the experiments are well performed and analyzed and it is an novel animal model for meibomian gland disease.

There are some minor criticisms.

1) The data shows that the ablation of cilia leads to an abnormal increase in lipid amount, which is not consistent with the most common mechanism for MGD, with a low delivery state characterized by gland obstruction. The underlying pathophysiology proposed to be terminal duct obstruction with decreased glandular secretion. The subtle change in meibum content is surprising.

We agree with the reviewer that a hallmark of MGD is a decreased secretion of lipids due to the obstruction of the MG central duct and, eventually, atrophy of the glands. In this study, we have examined the morphology of MG in *K14-Cre;Ift88^{fl/fl}* mutant mice lacking primary cilia in K14-expressing tissue. We observed an increase in meibum production and an expansion of the MG volume. However, we didn't detect any obstruction of the terminal central duct in the glands of the cilia mutant. The meibum was observed to exit the glands and form, at times, a milky deposition along the eyelid margins (Figure 1). This observation was clarified in the results section, *lines 139-140*. Our results suggest that the absence of primary cilia in the developing MG affects the amount of meibum production without altering its composition. As mentioned in the discussion of the manuscript, primary cilia seem to mainly control the cellular patterning of the glands during their development. Thus, we propose that rather than modeling MGD, the *K14-Cre;Ift88^{fl/fl}* mutant mouse models a possible avenue for therapeutic intervention by reversing atrophy during MGD. Our study aims to elucidate fundamental signaling cascades underlying gland expansion, ramification, and lipid production. Ultimately, this work may have identified potential therapeutic targets to combat MGD.

2) Dry eyes related to MGD may present with corneal epithelial defects and erosions. Some corneal surface evaluations/images would be helpful.

A previous publication from our laboratory addresses the effect of lack of primary cilium in corneal epithelial cells using an identical mouse model (Grisanti et al., 2016. *Development*. PMID: 27122169) to which we refer in several points of the manuscript ([ref 22] *lines 89, 121-122, 123-125*). We evaluated the corneal epithelium structure by histology and electron microscopy. We have shown that the ablation of primary cilia leads to a thicker corneal epithelium due to the decrease of Notch signaling, which induces an increase in proliferation

and vertical migration of basal corneal epithelial cells (Grisanti et al., 2016. Development. PMID: 27122169). However, because K14-Cre is expressed in the corneal epithelium, the conjunctiva and the lacrimal glands cilia are ablated in these tissues. Thus, it would be difficult to determine the direct effect of the defective MG on the cornea of a Keratin-14 cilia-deficient mouse. Unfortunately, to date, a specific Cre driver for MG is not available.

3) A number of MGD treatments have been used clinically (doxycycline, cyclosporine, etc.) does any of these reverse the phenotype observed?

We thank the reviewer for this information. Our present study, however, focuses on evaluating the effect of primary cilia deficiency on MG development. We are currently developing drug-delivery systems targeting MG using microgel formulations technology. We will certainly include the suggested drugs in these exciting future studies.

Minor

4) There is no “c” in Fig. 3 (duplicate e) and no P25 in Fig. 3h at line 166 and 173 in the manuscript, respectively. Please confirm the consistency of the manuscript and figures.

We apologize for the confusion. The labeling of the panels in fig 3 has been corrected and in Fig 3h, the mouse age indicated in the graph should read P25 as reported in the text. The graph has been corrected (P21 has been replaced by P25).

New Figure 3

5) Figure 6e should be log-scale or be two separate graphs (break in Y-axis seem to over-exaggerate the difference)

As suggested by reviewer 1, we split Fig 6e into two separate graphs: MG volume at P1, P3 and P4 are plotted in the same graph (Fig 6d) and MG volume at P8 is in a different graph (Fig 6f).

New Figure 6

Reviewer #2 (Remarks to the Author)

The manuscript by Portal et al. unveils a new role for primary cilia in Meibomian glands (MGs). The authors found that primary cilia regulate MG size and lipid content, but not lipid composition, possibly by regulating cellular patterning during early MG development. The study is interesting and in general carefully performed, with adequate controls and quantification of the results. However, the authors do not go deep enough into the mechanisms that can explain the observed phenotypes. In particular, they do not investigate any ciliary-dependent signaling pathways that can be involved in the regulation of the cellular patterning differences detected (and some of them are referred in the discussion). Therefore, the authors should investigate the expression of Hh (and/or Notch) signaling components in cKO MGs. But potentially more interesting would be to investigate planar cell polarity (PCP). In kidney tubules, PCP dysregulation caused by abnormal primary cilia

leads to enlarged ducts (and the formation of fluid-filled cysts that lead to cystic kidneys), which is somehow reminiscent of the phenotype that the authors describe. Thus, it would be valuable to investigate the expression and localization of PCP components.

We agree with the reviewer that uncovering the signaling(s) pathway(s) associated with the primary cilia mediating MG development and homeostasis would be very interesting. To address this rather vast question, we have examined the expression levels of several genes that are components or targets of signaling pathways known to be associated with the primary cilium in other tissues, including Hedgehog, Notch and Wnt pathways. We found that in adult mice, only the *Gli1* gene, a transcription factor and a target gene of the hedgehog (HH) signaling, appeared downregulated in dissected MG tissue from the cilia mutant *K14-Cre;ift88^{fl/fl}* when compared to control (**Fig. R1**). This result prompted us to investigate further the function of the hedgehog signaling on MG development.

To investigate the role of HH in MG development, we set out to delete *Smo*, encoding an obligatory component of the HH pathway, in epidermal cells. To do so, we have generated the conditional knock-out mice *K14-Cre;Smo^{lox/lox}* in which *Smo* is excised in all epidermal cells expressing K14 (Gritli-Linde et al., 2007). The majority of *K14-Cre;Smo^{lox/lox}* mutants did not survive adulthood and died between P18 and P23. MG morphogenesis was assessed by Oil Red O (ORO) staining of whole-mount eyelids. At P6 and P8, the number of ORO-stained MG in both the upper and lower eyelids of the mutant was significantly lower than that of the control (**Fig. R2A**). In the mutant, MG seem to develop mostly on the temporal side of the eyelids, although of smaller size than those of the control; moreover, the eyelashes were absent in the mutant eyelids (**Fig. R2A**). Longitudinal paraffin sections of the eyelids not only confirmed the remarkable reduction in the size of the MG of the mutant but also revealed the presence of eyelid closure, which is a requirement for normal MG development (Meng et al., 2014). Thus, indicating that the hylomorphic nature of the mutant MG is not due to defective eyelid closure (**Fig. 2B**).

Thus, the disruption of the HH pathway by ablating Smo, which is considered to be upstream of the cilium in the HH cascade, leads to an opposite effect compared to the MG phenotype of our cilia mutant. Currently, we are investigating whether the cilium is required for the integrity of the HH signaling cascade using mouse genetics [Portal, C., and Iomini, C. (2021). *Differential role of the Hedgehog pathway and the primary cilium in Meibomian gland development. Invest. Ophthalmol. Vis. Sci.* 62, 2050]. However, these studies are in progress and at this point, appear behind the scope of the present study. Prompted by the enlightening reviewer's comment, we are also following up on possible links between MG cilia and Planar Cell Polarity (PCP). Unfortunately, our attempts to localize PCP components in MG from mice at different ages using commercial and gifted antibodies directed to Vangl2, Frizzled3 and Celsr3 were inconclusive. PCP proteins have been proven difficult to localize by immunohistochemistry, this appears even more challenging in tissues that are particularly rich in lipids like the MG. Thus, we will pursue this interesting and intriguing avenue using mouse genetics tools in the near future. We thank the reviewer for this stimulating suggestion.

Other issues:

1) Arl13b is used as a ciliary marker and, indeed, it localizes to the ciliary membrane. However, another marker should be used to ascertain cilia abrogation, namely acetylated tubulin, which is a marker of the axoneme core (at least by immunofluorescence), to validate the results of Suppl. Fig. 1.

As suggested by the reviewer, we have further ascertained the ablation of primary cilia in the MG by using an antibody directed to acetylated tubulin (T6793, Sigma-Aldrich), which is highly enriched in cilia microtubules. However, this staining has been particularly challenging due to high background fluorescence, possibly due to the presence of numerous cytoplasmic acetylated microtubules in the developing MG. This set of experiments was added in Suppl. Fig. 1 (panels c and d) and the MATERIALS AND METHODS section has been updated (*lines 384-397*). The immunofluorescence experiments show that ARL13b-staining of cilia co-localize with acetylated tubulin-enriched structures. The number of double-stained structures is dramatically reduced or absent in the MG of the cKO mouse compared to the control. These results are consistent with ARL13B staining shown in Suppl. Fig. 1b.

New Supplement Figure 1

ARL13B mG/mT

2) In Fig. 3, the authors quantify the % of basal bodies associated with a primary cilium, which is not a standard measurement. Even though each cell should have only one basal body, the results should be displayed as % of ciliated cells. Since the authors have nuclear staining, this should be straightforward.

We agree that the % of ciliated cells is a more common way to express the number of cilia in a tissue than the % of basal bodies associated with a primary cilium. However, at the early stages of MG development, the density of cells is very high, which makes identifying individual nuclei on whole mount samples (P3, P6 and P8) challenging. Thus, instead of nuclei, we counted basal bodies, which were clearly visible as EGFP fluorescent dots on whole mount or frozen sections. The results were expressed as the percentage of basal bodies associated with cilia instead of the percentage of ciliated cells. However, following the reviewer's remark, we counted the number of nuclei and basal bodies on MG frozen sections at P25, when the array of cells is clearly distinguishable. We found that the number of nuclei is not significantly different from the number of basal bodies. Thus, the number of nuclei is similar to the number of basal bodies. Given this result, we modified the axis title of Fig 3g, 3h and Suppl Fig3 by "Number of ciliated cells (%)". We updated the method section related to the cilia localization to explain how the percentage of ciliated cells in MGs has been quantified (*lines 426-434*) and added the quantification of the nuclei and basal bodies in MGs at P25 as a supplemental figure (Supplement Fig 4).

New Supplement Figure 4

Nuclei and basal bodies were counted in 50x50 μm areas on P25 MG sections, and the quantification has been added in the manuscript as a supplement figure 4.

See new Figure 3 above

The axis title "Basal bodies associated with a primary cilium (%)" has been replaced by "Number of ciliated cells (%)" in panels g and h.

New Suppl Figure 3

The axis title "Basal bodies associated with a primary cilium (%)" has been replaced by "Number of ciliated cells (%)"

Minor issues:

3) In Fig. 6g, authors should label which schematic corresponds to control and cKO.

Following the reviewer's indication, we added labels to the schematic on Fig. 6h (which was previously Fig.6g).

See New Figure 6 above

Reviewer #3 (Remarks to the Author):

Portal et al, address the question of the role of primary cilia in MG development and function. To do so, they use convincing mouse genetic models. Based on these, they claim that primary cilia ablation affects the segregation of dividing cells compartments along the proximal-distal MG axis resulting in larger MG with increased lipid production.

Although we find the study globally well-conducted and of interest to the biologist community, and of potential relevance for DED, there are major points we would like to mention:

Major points:

1) Globally, quantifications are based on a low number of samples per condition. In Fig. 1d/ 2e/3g-h/4d-e-g-h/5f-g/6a-b-e-f, the authors should increase the number of samples to clarify the statistical significance.

We agree with the reviewer that, in some instances, the number of samples could be increased to produce optimal statistical analysis. Thus, we considerably expanded experimental replicates for each age and performed additional quantifications. This increase in number of samples per group has decidedly elevated the significance of statistical analyses throughout the study. However, it has been a lengthy process leading to go slightly beyond the mark of three months typically allowed for the revision.

Specifically, we added samples to reach a minimum of n=5 per group in Fig 1d (Number of MGs at P6 and MG area at P6) and a maximum of at least n=20 per group. We increased from n=3 to at least n=5 per group quantifications shown in Fig 5f (MG length increase at P3), Fig 5g (MG width increase at P3) and Fig 5l (Basal cells/central cells at P3).

We increased to n=5 samples for data shown in Fig 6a (MG central duct diameter increase at P6), and Fig 6b (MG central duct diameter increase at P21). Additional data points were also added in Fig 6e (MG volume) for each group to n=5. We did not include additional samples for quantification shown in Fig 2e, since here the number of replicates was already sizable (n=6/group).

To determine the optimal number of samples required for quantifications shown in Fig 3g-h (ciliated cells), we conducted a power analysis between the samples at P3 and P25 with G*Power 3.1.9.4. As shown below (G*Power analysis #1), 3 samples per group appear sufficient to obtain a significant result with alpha error 0.05, Wilcoxon-Mann-Whitney test (two groups) two tails. Thus, no additional replicates were added.

G*Power analysis #1

Fig 4d-e-g-h (EdU + TUNEL)

For quantifications of cell proliferation assessed by EdU staining (Fig 4d-e), 38 samples per group would be necessary to potentially get a significant result with alpha error 0.05, Wilcoxon-Mann-Whitney test (two groups) two tails (G*Power analysis #2). This number of mice per group appears unrealistic, especially considering that for this experiment we processed serial sections to count proliferating cells in the entirety of several MGs for each mouse. However, considering that 4 replicates are sufficient to get significant results while comparing the distal and proximal halves of MG in Ctrl mice at P4 [alpha error 0.05, Wilcoxon signed-rank test (matched pairs) two tails (G*Power analysis #3)], we used 4 mice per genotype for each time point in EdU experiments.

G*Power analysis #2

Parameter	Value
Test family	t tests
Statistical test	Means: Wilcoxon-Mann-Whitney test (two groups)
Type of power analysis	A priori: Compute required sample size - given α , power, and effect size
Input Parameters	
Tail(s)	Two
Parent distribution	Normal
Effect size d	0.8661112
α err prob	0.05
Power (1- β err prob)	0.95
Allocation ratio N2/N1	1
Output Parameters	
Noncentrality parameter δ	3.6892335
Critical t	1.9941516
Df	70.5746540
Sample size group 1	38
Sample size group 2	38
Total sample size	76
Actual power	0.9533866

G*Power analysis #2

G*Power analysis #3

Parameter	Value
Test family	t tests
Statistical test	Means: Wilcoxon signed-rank test (matched pairs)
Type of power analysis	A priori: Compute required sample size - given α , power, and effect size
Input Parameters	
Tail(s)	Two
Parent distribution	Normal
Effect size dz	3.4647425
α err prob	0.05
Power (1- β err prob)	0.95
Output Parameters	
Noncentrality parameter δ	6.7715276
Critical t	3.3007098
Df	2.8197186
Total sample size	4
Actual power	0.9830070

G*Power analysis #3

For the cell apoptosis assessed by TUNEL staining (Fig 4g-h), 197 samples per group would be necessary to potentially obtain a significant result with alpha error 0.05, Wilcoxon-Mann-Whitney test (two groups) two tails (G*Power analysis #4). This number of mice per group is not realistically attainable, thus we have maintained n=3.

G*Power analysis #4

For the quantification relative to the number of acini (Fig 6f), 763 samples per group would be necessary to potentially get a significant result with alpha error 0.05, Wilcoxon-Mann-Whitney test (two groups) two tails (G*Power analysis #5). Although these requirement appear out of reach, we have nevertheless added a replicate to obtain a total of n=5 (controls) and 4 (mutants).

G*Power analysis #5

2) The choice of statistical tests (Mann-Whitney, Anova, Wilcoxon,...) is not clear. Please explain in the statistics section the choices for using each test. The statistical test used should be mentioned in every figure legend.

For example, in Fig. 3h/4a-e-g-h/5i/6e-f/S2a-b/S3a: Student t-test should be used when the variances are similar and Mann-Whitney test when the variances are different. Use Fisher test (F test) to compare the variances of the two groups. Anova should be systematically used for multiple groups comparisons. In Fig. 5f-g/6a-b, use one sample t comparing with 100.

The paragraph detailing statistical analyses in the Material and methods has been rewritten to now include more information about the choices for using each test. As previously indicated, the statistical test is mentioned in the figure legend of each figure.

According to [Le Cessie et al., 2020. Eur J Endocrinol. PMID: 31910149], "When groups sizes are small (as a rule of thumb: below 25), the outcome variable should be normally distributed to use the t-test". However,

tests for normality do not have much power in small samples. Thus, for statistical analysis relative to this study, we opted for non-parametric tests.

In our case, ANOVA cannot be systematically used for multiple-group comparisons. ANOVA can be used for the comparison of multiple independent groups. However, in this study, groups are not always independent. In Fig3h, we are comparing the percentage of ciliated cells in the central duct and in the acini of the same MGs (paired samples). Similarly, in Fig4e, 4g and 4h, we are comparing the distal half and the proximal half of the same MGs (paired samples). Thus, ANOVA is not appropriate to compare these groups. In Fig3g and Fig6d, which compare multiple independent groups, we used Kruskal-Wallis test (non-parametric equivalent of the ANOVA test).

In Fig 5f-g/6a-b, we compared 5 mutants to 5 controls. The data of each mutant was normalized to the value of its littermate control, and thus all the control values are 100%.

Minor points:

3) The order of the figures and lettering. The authors need to revise these, especially for Figure 4, as it makes the manuscript difficult to read.

Thanks to the feedback of the reviewer, we have carefully re-examined the order and the lettering of all figures. For Figure 1, we inverted Fig1c and Fig1d, so now all the panels of Fig1 are mentioned in the manuscript following a numbered order. The figure legend has been updated accordingly. The legend of Figure 1 is now (*lines 688-699*): **“Fig. 1: Primary cilium ablation leads to larger MGs. (a)** Representative pictures of control and cKO adult (6 months) eyes. Arrow indicates a white deposit, only observed in cKO mice. **(b)** Representative images of tarsal plates stained with ORO at P6 and P8. Boxed regions indicate the areas shown at higher magnification. Scale bar; 200 μ m; N, nasal; T, temporal. **(c)** Number of MGs and MG size were quantified at P6 and P8 (n=20 controls and 5 cKO mice at P6; n=13 controls and 9 cKO mice at P8). Per mouse, MG area was determined by averaging the MG area of all individual MGs in the upper and lower eyelids. Data were presented as mean \pm SD. Statistical significance was assessed using Mann Whitney test. ns, non-significant, $P \geq 0.05$. **(d)** Representative images of tarsal plates stained with ORO at P21. Boxed regions indicate the areas shown at higher magnification. Scale bar; 200 μ m; N, nasal; T, temporal. **(e)** Representative images of control and cKO MGs stained with HE at P6 and P21. Boxed regions indicate the areas shown at higher magnification. Scale bar; 200 μ m.”.

New Figure 1

In the manuscript, we refer more often to panels of Figure 4 and hope that this part of the result section appears easier to follow (*lines 185-223*).

4) Fig. 2e: legend is unclear. Missing statistics.

We apologize for this omission and thank the reviewers for the accurate remark, we added the statistics on the graph of Fig 2e. In the figure legend, we removed the sentence “cKO corresponds to *K14-Cre;ift88^{fl/fl}*, ...remaining genetic combinations” (*lines 702-703*) that was confusing and not necessary since the heterozygous mice were included in the control group, similar to the rest of the manuscript.

New Figure 2

5) Figure 3: Fig 3b not mentioned in the text. In the figure “e” instead of “c” and no scale bars in 3c and 3d. 3h legend is unclear.

Our apologies to the reviewers for this omission and confusion. Fig 3b is now mentioned in the manuscript (*lines 164-166*): “Throughout this study, we analyzed glands preferentially located at a similar middle position in the tarsal plate, as illustrated in **Fig. 3b**, in mutant and control mice.”.

The lettering of Fig 3c has been corrected and scale bars were added in Fig 3c and 3d.

See New Figure 3 above

The legend of Fig 3h has been rewritten to emphasize that the number of ciliated cells was quantified in two different glands regions (acini vs. duct). The new Fig 3h legend is as follows (*lines 741-746*): “(h) Spatial distribution of ciliated cells within MGs at P12 and P25. The percentage of ciliated cells was quantified specifically in the acini and in the central duct of MGs at P12 and P25 (n=3 for each age). Data were presented as mean \pm SD. Statistical significance was assessed using Mann Whitney test (Ctrl vs. cKO) and Wilcoxon signed rank test (acini vs. duct). ns, non-significant, $P \geq 0.05$.”.

6) Fig. 4d-e: please add the name of the measured areas on the graphs.

In Fig 4d-e, cell proliferation has been quantified in the full MGs. This information has been added to the axis title of both graphs.

New Figure 4

REVIEWERS' COMMENTS:

Reviewer #1 (Remarks to the Author):

The authors have addressed this reviewer's concerns. There is no further outstanding issues.

Reviewer #2 (Remarks to the Author):

The authors addressed thoroughly the reviewers' comments and significantly improved the manuscript. In the case of Figures R1 and 2, the authors should at least refer them in the discussion. Indeed, the manuscript would benefit from the mechanistic insight provided by the data displayed in these Figures and their inclusion in the manuscript.

Reviewer #3 (Remarks to the Author):

Overall, the majority of our concerns were adequately addressed.